# Extracellular Vesicles Secreted by Pre-Hatching Bovine Embryos Produced In Vitro and In Vivo Alter the Expression of IFNtau-Stimulated Genes in Bovine Endometrial Cells

**DOI:** 10.3390/ijms24087438

**Published:** 2023-04-18

**Authors:** Constanza Aguilera, Alejandra Estela Velásquez, Miguel Angel Gutierrez-Reinoso, Yat Sen Wong, Barbara Melo-Baez, Joel Cabezas, Diego Caamaño, Felipe Navarrete, Daniela Rojas, Gonzalo Riadi, Fidel Ovidio Castro, Llretny Rodriguez-Alvarez

**Affiliations:** 1Laboratory of Animal Biotechnology, Department of Animal Science, Faculty of Veterinary Sciences, Universidad de Concepción, Av. Vicente Mendez 595, Chillan 3780000, Chile; consaguilera@udec.cl (C.A.);; 2ANID-Millennium Science Initiative Program Millennium Nucleus of Ion Channels-Associated Diseases (MiNICAD), Center for Bioinformatics, Simulation and Modeling, CBSM, Department of Bioinformatics, Faculty of Engineering, Campus Talca, University of Talca, Talca 3460000, Chile

**Keywords:** extracellular vesicles, embryo-maternal communication, interferon tau, bovine embryos, endometrial bovine cells

## Abstract

The embryo-maternal interaction occurs during the early stages of embryo development and is essential for the implantation and full-term development of the embryo. In bovines, the secretion of interferon Tau (IFNT) during elongation is the main signal for pregnancy recognition, but its expression starts around the blastocyst stage. Embryos release extracellular vesicles (EVs) as an alternative mechanism of embryo-maternal communication. The aim of the study was to determine whether EVs secreted by bovine embryos during blastulation (D5-D7) could induce transcriptomic modifications, activating IFNT signaling in endometrial cells. Additionally, it aims to assess whether the EVs secreted by embryos produced in vivo (EVs-IVV) or in vitro (EVs-IVP) have different effects on the transcriptomic profiles of the endometrial cells. In vitro- and in vivo-produced bovine morulae were selected and individually cultured for 48 h to collect embryonic EVs (E-EVs) secreted during blastulation. E-EVs stained with PKH67 were added to in vitro-cultured bovine endometrial cells to assess EV internalization. The effect of EVs on the transcriptomic profile of endometrial cells was determined by RNA sequencing. EVs from both types of embryos induced several classical and non-classical IFNT-stimulated genes (ISGs) and other pathways related to endometrial function in epithelial endometrial cells. Higher numbers of differentially expressed genes (3552) were induced by EVs released by IVP embryos compared to EVs from IVV (1838). Gene ontology analysis showed that EVs-IVP/IVV induced the upregulation of the extracellular exosome pathway, the cellular response to stimulus, and the protein modification processes. This work provides evidence regarding the effect of embryo origin (in vivo or in vitro) on the early embryo-maternal interaction mediated by extracellular vesicles.

## 1. Introduction

The embryo-maternal interaction occurs during the early stages of development and is essential for the implantation and full-term development of an embryo [1,2,3]. The bovine embryo reaches the uterus on Day 4 during the stage where it consists of 16 cells or morula. From this moment on, the embryo interacts with the maternal side in a paracrine manner until the attachment period begins at around Day 19 of development [3]. In cattle, the main signal for pregnancy recognition is IFNT, which is produced by the embryo during the blastocyst stage [4,5]. IFNT secretion increases during elongation, acting on the endometrial cells to indirectly prevent luteolysis [6]. Although IFNT secretion peaks at Day 18 of pregnancy, several reports indicate that early IFNT secretion by the blastocysts of cattle induces the upregulation of interferon-stimulated genes (ISGs) in endometrial cells, whether in an in vivo or an in vitro model [7,8,9,10,11,12].

For decades it was accepted that, before the blastocyst stage, endometrial receptivity is induced and maintained by the maternal hormonal state after ovulation [13]. However, during early development, the embryo produces signals that alter the maternal cells. Studies on mice and pigs show that the presence of multiple embryos generates signals that change the transcriptomics and the secretomics of oviductal cells [14,15]. However, in uniparous species, the signals and the maternal effect induced by a single embryo are difficult to detect. In a report on cattle, when the embryonic signal increased due to the presence of several embryos in the oviduct, a transcriptomic change was detected in the oviductal cells [16]. Deciphering what signals are produced by the embryo and how important they are in preparing the maternal side for implantation is of pivotal importance.

Recently, the role of extracellular vesicles in mediating embryo maternal communication has been described [2,17,18,19]. EVs are nanoparticles surrounded by a lipid bilayer that are involved in intercellular communication. EVs induce functional modifications in cells that are capable of internalizing them. They carry bioactive molecules such as proteins, lipids, mRNA, and miRNA and play a role in intercellular communication through their cargo [20,21,22]. EVs are involved in different biological processes and participate in the cellular communication of different cell types [22,23,24,25].

The role of EVs in the reproductive process has been demonstrated [18]. The EVs contents of oviductal and uterine fluids differ between pregnant and non-pregnant cows [26]. Maternal-derived EVs are internalized by embryos cultured in vitro, modifying their quality, competence, and cryotolerance [2,27,28]. In humans, EVs secreted by trophoblast spheroids alter the transcriptomic profile of endometrial cells in the culture [29]. EVs derived from elongated sheep conceptuses are internalized by the uterine luminal and glandular epithelial cells [30]. Additionally, in sheep conceptuses, derived EVs contain IFNT and induce pathways associated with IFNT signaling in endometrial cells [31]. Bridi [32] found that bovine embryos secreted EVs containing miRNAs from Day 7 up to Day 9. These EVs can be implicated in the endometrial modifications required for implantation. EVs could provide an alternative mechanism to soluble INFT in ruminants. However, to the best of our knowledge, it is unknown whether the population of EVs secreted by early embryos at Day 5 of development interact with the endometrium preparing it for pregnancy recognition. Furthermore, it is known that IVP and IVV (IVP: in vitro-produced; IVV: in vivo-produced) embryos differ in their developmental competence, gene expression pattern, and molecular secretion [33,34,35]. These differences can influence the embryo’s signaling ability to modify the endometrium, which could explain the difference in implantation rate between these embryos [36,37,38]. For instance, IVP ovine embryos inhibit the expression of endometrial proteins related to mitochondrial metabolism, glycolysis, and cellular proliferation, compared to those produced in vivo [39]. Additionally, bovine IVP embryos induce different gene expressions in endometrial transcriptome and activate different biological pathways compared to IVV embryos [37]. Moreover, it has been reported that there were changes in the miRNA cargo of EVs released by embryos produced in vitro compared to those produced in vivo [32].

This work studies the effect of EVs secreted by cattle embryos during blastulation on endometrial cells upon in vitro co-culture. The main goal was to determine how early the embryo interacts with the maternal environment, causing the activation of IFNT-associated pathways. Additionally, the effect of embryo origin, in vivo or in vitro, on the transcriptomic profile of the endometrial cells was evaluated.

## 2. Results

### 2.1. Characterization of Nanoparticles Collected from Embryo Culture Media

The nanoparticle distribution was evaluated to determine the concentration and size (mean and mode) of particles isolated in pools of culture medium from bovine embryos produced in vitro and in vivo. The statistical descriptors and graphic representations of the data are represented in Figure 1 and Table 1. The isolated nanoparticles from culture media of in vivo- and in vitro-derived embryos as well as those from bovine blood serum (positive control) were positive for the EVs markers CD9, CD63, CD81, and CD40 (Table 2) [40]. However, negative controls (SOFdep: depleted oviductal synthetic fluid) showed low positivity compared to samples (Table 2). Furthermore, nanoparticles were visualized by TEM, showing a classical EV morphology with a round, cuboidal shape (Figure 2).

### 2.2. Internalization of Extracellular Vesicles

Bovine endometrial epithelial cells were incubated with PHK67-stained EVs secreted during the blastulation of embryos produced in vivo or in vitro. Green spots were observed in the cell cytoplasm when EVs from in vivo or in vitro embryos were added to the culture. However, no fluorescent signal was observed when PBS/PHK67 was added to the cells (Figure 3).

### 2.3. Transcriptomic Analysis in Endometrial Epithelial Cells

The effect of embryo-derived EVs on the mRNA profile of endometrial epithelial cells was evaluated using mRNA sequencing. The number of sequence reads per sample and the mapping rates are provided in Appendix A. Data from RNAseq were uploaded to NCBI’s Sequence Read Archive (SRA) with accession code PRJNA931759. 

Low variability was observed for the experimental groups compared to the control group, which is evidenced in the analysis of principal components (PCA) (Appendix A). However, it is possible to identify a cluster for each experimental group, indicating a differential profile for the mRNA of vesicles from IVV and IVP embryos.

A total of 14,987 genes were detected across samples. Differential expression analysis was performed, and genes were considered differentially expressed when log2 fold change (log2FC) > 0.5 or <−0.5 and FDR ˂ 0.05 (Figure 4a,b). The effect of EVs from in vivo-produced or in vitro-produced embryos was independently evaluated by comparing them with the PBS treatment (negative control). The EVs-IVP samples induced a higher number of differentially expressed genes (3552) compared to the EVs secreted by in vivo-derived embryos (1839) (Figure 4c,d). In both comparisons, the number of upregulated genes was higher, but this effect was even greater when the cells were exposed to EVs from in vitro embryos (Figure 4c).

An overlapping analysis showed that 29.6% of genes were deregulated when cells were incubated with EVs, independent of the embryo origin, while 55.8 and 14.6% of deregulated genes were included exclusively in the EVs-IVP/control and EVs-IVV/control groups, respectively (Figure 4d). Among the deregulated genes, several classical and non-classical IFN-stimulated genes (ISGs), as well as genes related to endometrial function, were affected in the same manner (up or down) in cells exposed to EVs from IVV and IVP. However, there is a tendency, shown by the gene expression fold change, for EVs from IVP embryos to have a stronger effect (inducing or repressing gene expression) on the cells. This tendency is also visualized in a heat map figure performed using deregulated genes listed in Table 3 (Figure 5). Deregulated genes were organized into three clusters, showing clear differences between the control and experimental groups. However, a differential pattern is also observed between EVs-IVP and EVs-IVV groups for Clusters 1 and 2 that includes classical and non-classical ISG and genes associated with endometrial function.

From the previous analysis, it was detected that EVs have a different effect on endometrial cells depending to the origin of the embryo. The impact of this effect was evaluated by a differential gene expression analysis of both the EVs-IVV and EVs-IVP groups, which detected 3968 deregulated genes (Figure 6). This analysis confirmed that EVs released by IVP embryos had a greater effect on the induction of gene expression. Of the deregulated genes, 24 were identified as being involved in the IFNT pathway or endometrial functions (Table 4).

### 2.4. Validation of RNA Seq through Quantitative Reverse Transcription PCR Analysis

To validate the results of the RNAseq, the relative expression levels of 13 genes were determined by real-time PCR. Using the RNAseq data, these genes were found to deregulate when endometrial cells were exposed to EVs from IVV and/or IVP embryos. These genes were also selected based on their biological relevance (downregulated: *ESR1*, *HPGD*, *MMP14*, *PTGES*; upregulated: *CST6*, *CTSL*, *MUC1*, *MX1*, *MX2*, *OAS1Y*, *OXTR*, *WNT7A*, *ISG15*. All of the genes except MX2 in the EVs-IVP group demonstrated similar behavior (up or down) when comparing the expression pattern obtained by RNAseq and PCR (Figure 7). A high correlation between the expression level determined by RNAseq and PCR was determined using Pearson correlation analysis (R: 0.88, *p* = 0.00018 in EVs-IVP and R: 0.73, *p* = 0.005 in EVs-IVV) (Figure 7). PCR analysis confirmed the upregulation of interferon-induced genes (*CST6*, *CTSL*, *MX1*, *ISG15*, *MX2*, *WNT7A*, and *OAS1Y*) in cells incubated with EVs produced by IVP and IVV embryos, and the greater effect of EVs secreted by in vitro derived embryos was also confirmed.

### 2.5. Gene Ontology Analysis

A gene ontology analysis (G.O) using the differentially expressed genes (DEGs) between EVs-IVV/control and EVs-IVP/control was performed to determine other affected pathways. The measure of magnitude was assessed using –log10 (*p*-value) to determine statistical differences between the groups (Appendix A). The significantly involved pathways were classified as biological processes (BPs), cellular components (CCs), or molecular functions (MFs). All EVs induced several pathways including the cellular component of the extracellular exosomes, independent of an EV’s origin. Moreover, the results indicated that EVs-IVV significantly induced different biological process (BP) pathways linked to the protein modification process, the regulation of gene expression, embryo development, and blood vessel development (Appendix A). The number of biological processes activated by EVs-IVV/control was higher than those induced by EVs-IVP/control. Otherwise, the more represented BPs in EVs-IVP/control were gene expression, cellular response to DNA damage, and response to UV stimulus (Appendix A).

## 3. Discussion

In this study, it was demonstrated that EVs secreted during blastulation by pre-implantation bovine embryos produced in vitro and in vivo are internalized by endometrial cells. After 24 h of in vitro incubation, PHK67-stained EVs were detected in the cytoplasm of the cells. Extracellular vesicles are a mechanism of cell–cell communication because they carry signaling molecules that alter the physiological state of the recipient cells. There is evidence that EVs are a mechanism for embryo–maternal communication, which seems critical during pregnancy recognition before the main signal is induced by the embryo. For instance, in humans, embryo/trophoblast-derived EVs are internalized by primary endometrial epithelial and stromal cells and alter the gene expression pattern in a specific way [29,62]. In this work, we aimed to elucidate whether bovine embryonic EVs released during the early stages of development affect pathways related to endometrial response during pregnancy, and how the origin of the embryo can influence this effect.

The addition of embryo-derived EVs to the cell culture modified the mRNA expression level of classical and non-classical interferon-induced genes. In bovines, IFNT is the main embryonic signal used to prolong the CL lifespan and therefore the secretion of progesterone required to maintain endometrial conditions for implantation [6]. IFNT stimulates the expression of genes (ISGs) in endometrial cells involved in cell differentiation and conceptus development, which leads to implantation [63,64]. In cattle, the peak of IFNT secretion takes place at Day 16 of development [65,66]. However, in vivo-produced bovine embryos begin to secrete IFNT at the blastocyst stage, which stimulates classic ISGs in endometrial cells in vivo and in vitro, and modulates the immune response in the uterus [7,8,9,67].

Here, classical interferon-induced genes (*OAS1Y*, *MX1*, *MX2*, and *ISG15*) were upregulated in the endometrial cells when EVs released by embryos produced in vitro and in vivo were added to the culture medium. The Jak/Stat pathway activates the interferon-stimulated response elements (ISRE) that induce the expression of ISGs [6]. However, in this work, the components of the JAK-STAT signaling system, *STAT1*, *STAT2*, and *IRF9*, were detected in all groups, and their expression levels were not affected by embryo-derived EVs. However, alternative pathways of ISGs induction have been described. For instance, in ovines, there is a lack of JAK-STAT pathway components in the epithelial luminal cells in the endometrium, and it seems that the MAPK and PI3K pathways activate ISGs [51,68]. Other genes, such as *IRF6*, an interferon regulator factor, were upregulated by EVs released by IVV and IVP embryos. This factor is present in epithelial endometrial ovine cells and activates ISRE-containing promoters, which leads to the expression of ISGs [46]. This finding suggests an alternative mechanism to activating the ISRE-containing promoter other than through the classic pathway.

Genes related to endometrial function, such as *PTGS1* and *PTGS2*, were upregulated in the cells exposed to EVs released by IVP and IVV embryos. At least in ovines, IFNT increases the endometrial production of PTGSs and the amount of prostaglandins in the uterus [53]. In this species, the *PTGS2* expression in the endometrium and the production of prostaglandins by the embryonic trophoblast is required for conceptus elongation [53,69]. This data may suggest that EVs released by the early embryo play a similar or complementary roll to IFNT. The mRNA for *AKR1C4*, the enzyme in charge of PGF2α production [50]; *MUC1*, a gene coding for a protein associated with endometrium receptivity; *OXTR* (oxytocin receptor); and *PAQR8* (progestin and adipoQ receptor family member 8) were upregulated in the endometrial cells due to the addition of EVs. In bovines, MUC1 is upregulated at Day 7 of pregnancy [8,70], and its antiadhesive properties could help the embryo to move through the reproductive tract prior to implantation. *AKR1C4* was more expressed by EVs from IVP embryos. This enzyme is responsible for the production of PGF2α, which has a potent luteolytic effect [50]. Zhou et al. [71] mention that porcine blastocysts produced by somatic cell chromatin transfer and parthenogenetic activation are less competent, and they also induced greater expression of *AKR1C4* in the endometrium compared to in vivo ones. In agreement with this, in an independent study, it was found that bovine blastocysts produced in vitro release EVs containing miR-221 [72], which is involved in the induction of AKR1C4 as a protein and mRNA in cultured cells [73].

During pregnancy, the progesterone secreted by the corpus luteum suppresses the expression of *ESR1* and *OXTR* [74]. However, in this work, the OXTR expression was upregulated by the EVs but to a greater degree by EVs derived from IVP embryos. This is in accordance with an in vitro experiment where *OXTR* mRNA levels increased in endometrial explants exposed to pre-implantation embryos [32]. This discrepancy between in vivo and in vitro experiments might be due to the absence of progesterone. On the other hand, *PAQR8* was upregulated in the endometrial cells exposed to embryo-derived EVs. This receptor is involved in the non-genomic mechanism of progesterone response, which is faster than the genomic action of the progesterone receptor (PGR), probably because it is present in plasma membrane and can be activated in minutes once it is exposed to progesterone [75,76]. In humans, the decreased expression of PAQR8 is associated with resistance to progesterone in patients suffering from endometriosis [77]. This could suggest that EVs are a mechanism to sensitize the endometrial cells to the action of progesterone through its membrane receptor.

EVs induced a higher number of upregulated genes in the endometrial cells; however, several genes related to endometrial remodeling have a tendency to be downregulated; among them is the matrix remodeling gene *MMP19*. This result agrees with other reports where *MMP14* and *MMP19* were downregulated in pregnant cows at Day 7, probably because the endometrium remodeling is required later, at the end of the peri-implantation period [8].

With respect to prostaglandin E synthase (PTGES), there is a tendency in the RNAseq towards the downregulation of this enzyme; however, EVs-IVV induced more expression compared to EVs-IVP. The expression of PGE2 is crucial for the protection of corpus luteum and the maintenance of pregnancy [50]. Most likely, this lack of induction of PGE2 by the EVs from in vitro-produced embryos would affect their competence compared to in vivo ones.

The results show that pre-implantation embryos released extracellular vesicles that interacted with the maternal side by changing (inducing or repressing) the gene expression pattern of endometrial cells. However, it seems that the origin of the embryo impacts the characteristics of released EVs and consequently modulates their effect on maternal cells. The competence and characteristics of the embryo can be sensed by the endometrium, and the quality, size, and sex of the conceptus can induce a different transcriptomic profile in the endometrial cells [36,38,78]. It is known that IVV embryos are more competent than their IVP counterparts because in vitro-produced embryos are subjected to a high level of stress [37,71,79].

Independently of IFNT, the endometrium can respond differentially in the presence of an embryo produced in vitro or in vivo [37,38]. Several studies indicate that IFN production is higher in embryos produced in vitro, but there are results that contradict this statement [33,80]. Additionally, although the number of reports is limited, there is evidence showing that the size, concentration, and molecular content of embryonic EVs are also modified by the embryo competence, its origin, and the pre-implantation stage [81,82,83,84]. A recent study showed differences in miRNA cargo in EVs from in vitro and in vivo preimplantation bovine embryos [32]. Moreover, the same study demonstrated that the origin of the embryo determined the communication with the endometrium and corpus luteum. The above data could explain the differential gene expression pattern in the endometrial cells incubated with EVs from IVV and IVP embryos.

In this study, several genes were differentially regulated when the endometrial cells were exposed to EVs from IVV or IVP embryos. A higher number of genes and pathways were modified by the action of EVs secreted by IVP embryos, including genes related to secretory activity (*LTF*, *MCOLN3*, and *PIP*), glucose transporters (*SLC2A1*), growth factor signaling (*IGF2BP3*), endometrium receptivity (*MUC1*), and IFNT response (*CTSL*, *HSD11B1*, *IRF6*, and *SPP1*). In an in vivo experiment on pregnant cows, the expression of glandular secretion and solute transport was reported in endometrium exposed to bovine blastocyst, while the growth factors were downregulated [8]. The estrogen receptor alpha (*ESR1*) was upregulated by EVs from IVP embryos, but had the tendency to be downregulated by EVs released by in vivo-produced embryos. The mRNA level of *ESR1* is high on Days 5 and 7 after ovulation in cattle, but decreases around the time of implantation [85,86]. In ruminants, IFNT silences the expression of ESR1 and, in turn, oxytocin receptors to prevent the activation of luteolytic mechanisms [87]. In this work, it was found that OXTR increased, but its expression was higher in cells cultured with EVs from IVP embryos. This could be an effect of the different expression of *ESR1*, which is upregulated only by EVs-IVP. Such differences in the endometrial cells’ response induced by EVs from in vivo- and in vitro-produced embryos might be explained by a different molecular content in those vesicles, which could be a consequence of embryo competence.

In this work, the contents of embryo-derived EVs was not analyzed. However, in an independent experiment conducted by our research group, several miRNAs that are involved in the regulation of the aforementioned genes were detected in EVs released by bovine embryos [72,81]. For instance, miR29 family genes miR-221 and miR143-3p were detected in embryonic EVs and were related to the regulation of PTGS2, AKR1C4, and progesterone receptors including PAQR8 [72,73,88]. The modification of expression pattern of genes related to the IFNT pathway suggest its presence or that of other related molecules in the EVs.

Finally, with respect to the GO analysis, the results indicate that different pathways are activated by EVs-IVP and EVs-IVV. Our results indicated that embryo-derived EVs modified several pathways in the endometrial cells, including components of the exosome biogenesis. However, the effect of EVs controlling different pathways is determined by the origin of the embryo. These results are in concordance with the work from Mansouri-Attia et al. [37], where GO analysis showed that the pathways activated in endometrium explants are dependent on the technology used for embryo production.

## 4. Materials and Methods

The procedures were carried out under the approval of the Ethics Committee of the Faculty of Veterinary Sciences, University of Concepcion (permit no. CBE-17-2017).

### 4.1. Experimental Design

EVs secreted during blastulation by in vivo-produced (EVs-IVV) or in vitro-produced (EVs-IVP) embryos were isolated, labeled, and added (Time 0) to in vitro-cultured epithelial cattle endometrial cells. EV uptake was evaluated 24 h after incubation by visualization using PKH fluorescence staining. Cells were kept in culture for another 24 h and then harvested for gene expression analysis. PBS subjected to similar treatment to embryo-derived EVs was used as a negative control. The experiment was performed in triplicates for each experimental group: EVs from in vivo embryos, EVs from in vitro embryos, and PBS (Figure 8).

### 4.2. In Vitro Embryo Production

Bovine ovaries were collected from the local abattoir and transported to the laboratory in ~37 °C saline (0.9% *w*/*v*). The cumulus-oocytes complexes (COCs) were aspirated from antral follicles and selected using a stereo microscope. The COCs were matured in vitro (IVM) in groups (25–30 COCs) in a 4-well culture dish (Nunc, Thermo Scientific, Waltham, MA, USA) for 24 h in TCM 199 Earle’s buffered medium supplemented with glutamine (0.6 mM), pyruvate (0.2 mM), FSH/LH (0.2 U/mL), estradiol (1 µg/mL), gentamicin (50 µg/mL), EGF (epidermal growth factor; 10 ng/mL), and 10% (*v*/*v*) of FBS (fetal bovine serum). The COCs were incubated at 38.5 °C in an atmosphere of 5% CO_2_ in air. The matured COCs were fertilized in vitro (IVF) using thawed frozen commercial semen (Semex, Madison, WI, USA). Sperm quality was assessed in an independent experiment demonstrating a good embryo production rate using standard IVF procedures from our laboratory [89]. Motile sperm were selected using the Percoll method. For this, a centrifugation gradient was performed through different Percoll percentages (45%/90%) at 4500 RPM for 6 min. The sperm pellet was washed using 500 µL of IVF medium (TALP-IVF) supplemented with 0.01 mg/mL of heparin, 2 mM of pyruvate, 50 µg/mL of gentamicin, and 6 mg/mL of BSA (bovine serum albumin) and centrifuged at 3000 RPM for 3 min. Fertilization was carried out in groups of 25–30 COCs with a sperm concentration of 1 × 10^6^/mL. Incubation was performed in 4-well culture dishes with 500 µL of IVF medium per well and in an atmosphere of 5% CO_2_ in air. After 18–20 h of IVF, the presumptive zygotes were denuded of remaining cumulus cells by mechanical vortex (4 min) in TCM 199-hepes with 0.3 mg/mL of hyaluronidase. Zygotes were cultured in 4-well culture dishes (25–30 zygotes per well) with 500 µL of SOF (oviductal synthetic fluid) supplemented with 0.37 mM of trisodium citrate, 2.77 mM of myoinositol, 10 ng/mL of EGF, 2% of FBS (fetal bovine serum), and 3 mg/mL of essentially fatty acid-free BSA for 5 days at 38.5 °C under an atmosphere of 5% CO_2_, 5% O_2_, and 90% N_2_. On Day 5, embryo development was evaluated; morulae were collected and transferred to individual culture drops of 40 μL of EVs-depleted SOF medium (SOFdep) until Day 7. SOFdep was produced through the ultrafiltration (centrifugal filter devices 100 kDa, Amicon, Merck, Darmstadt, Germany) of supplemented SOF medium for 15 min at 1660× *g* at 4 °C.

### 4.3. In Vivo Embryo Production

In vivo-produced embryos were obtained using superovulation procedures prior to artificial insemination (AI). Ten Red Angus heifers were treated using the following protocol: on Day 0, an intravaginal P4 device (CIDR 1.38 gr, Zoetis, Parsippany-Troy Hills, NJ, USA) was applied, and 2.5 mg intramuscular (im) Estradiol Benzoate (Syntex, Zoetis) (E2B) and 100 mg of progesterone (P4) (Gestavec, Vecol, Freienbach, Switzerland) (im) were administered. From Day 4 until Day 7, decreasing doses of bscFSH-r (Cebitropin, Concepción, Chile) were administrated every 24 h (65–55–45–35 µg). On Day 6, two PGF2α doses were administrated 12 h apart (im; 500 µg D-cloprostenol (Zoetis) each). The CIDR (Pfizer, S. A., New York, NY, USA) device was removed on Day 7. On Day 8, 0.021 mg of GnRh (Virbac, Carros, France) was applied. All treatments were performed at 7:00 am. Then, AI was performed 12 h after heat detection. The second AI was performed 12 h after the first insemination. Donors were inseminated with the same semen used for IVF. Embryo collection was conducted 5 days after heat detection to collect structures at the morulae stage. Embryo recovery was performed by nonsurgical flushing of the uterine horns with 500 mL of PBS supplemented with 1% of FBS per horn. Collected structures were classified following the criteria from the IETS manual [90]. Morulae were selected and individually cultured for 48 h in similar conditions to the IVP ones.

### 4.4. EV Isolation from Embryo Culture Medium

In vivo- and in vitro-produced morulae were selected and cultured individually for 48 h (until Day 7; blastocyst stage). Only culture medium from grade I and II expanded blastocysts derived from in vitro or in vivo morulae were collected based on their morphology to define the quality and developmental stage [90]. Culture media from blastocysts were pooled for each experimental group (in vivo: n = 50 or in vitro: n = 95) for EV isolation. EVs were isolated using the exosome isolation kit “ExoLutE^®^ Conditioned Medium” according to the manufacturer’s instructions (Rosetta Exosome Inc., Seongnam-si, Republic of Korea). This Exosome Isolation Kit uses a method based on size-exclusion chromatography, which provides a high concentration of intact EVs with high purification. In brief, a pre-clearing step was conducted by centrifuging the sample at 400 g for 10 min, followed by another centrifugation of the supernatant at 2000 g for 20 min. The supernatant was recovered and subjected to an EV enrichment procedure. The EVs were recovered and resuspended in solution R to be applied into the column L. Recovered EVs were purified by loading 50 µL of the sample in column S and centrifuged at 700 g for 5 min. The recovered volume containing the EVs was stored at −80 °C for later use.

### 4.5. EVs Characterization

#### 4.5.1. Nanoparticle Tracking Analysis

The size and concentration of EVs from both in vitro- and in vivo-derived embryos were determined by nanoparticle tracking analysis (NTA) using the NanoSight NS300 (Malvern Instruments Ltd., Malvern, UK). This equipment is suitable for determining the size of particles between 10 to 2000 nm. For the readings, the particle concentration was set to 20–100 particles per frame. Filtered PBS was used as a negative control. The individual samples were loaded into 1-mL tuberculin syringes and connected to the syringe pump for NanoSight (Malvern Instruments Ltd., Malvern, UK), allowing the analysis of samples at a constant flow rate. Each sample was analyzed in triplicate with the same camera configuration, an acquisition time of 60 s, and the detection threshold setting at level 8. The measurement was performed using the 488 nm laser. The data were captured and analyzed using the incorporated software (NTA version 3.2 Dev Build 3.2.16, Malvern Instruments Ltd., Malvern, UK).

#### 4.5.2. Analysis of Surface Markers by Flow Cytometry

The evaluation of specific EV markers (CD9, CD63, CD81, and CD40) defined by the ISEV (International Society of Extracellular Vesicles) was performed using flow cytometry [40]. The procedure was carried out as described by Mellisho et al. [82] for EVs derived from pre-implantation bovine embryos. To this end, 4 × 10^8^ particles from each experimental group were mixed with 1.25 × 10^5^ of 4 μm-diameter latex beads (Life Technologies, Santiago, Chile) in a final volume of 100 μL of PBS and incubated for 18 h at 4 °C. Subsequently, 22 µL of 1 M glycine were added to block the unbound sites of the latex beads, gently mixed, and incubated for 45 min at room temperature (RT). The EV/bead complex was washed twice with 1 mL of PBS supplemented with 0.5% BSA by centrifuging at 1500× *g* for 3 min at RT. A measure of 100 μL of EV/bead complexes were recovered and incubated for 1 h at RT with antibodies previously tested in EVs derived from bovine samples [72]: against CD63 (FITC-conjugated; Abcam, ab18235), CD9 (FITC-conjugated; Abcam, ab34162), CD81 (PE-conjugated; Abcam ab81436), or CD40L (PE/Cy5^®^-conjugated; Abcam ab25044). For each antibody, a control reaction was set (antibodies incubated with latex beads but not with the EVs, 1 h at RT). Synthetic oviductal fluid medium (SOF) depleted of EVs was used as an experimental negative control, while EVs recovered from bovine blood serum and conjugated with latex beads, as described above, were used as positive controls. To perform the analysis, 100 µL of the EV/bead complexes were resuspended with 400 µL of focus fluid buffer for flow cytometry using the Attune ™ NxT Flow Cytometer (Life Technologies, Inc., Frederick, MD, USA).

#### 4.5.3. Transmission Electron Microscopy Analysis

A volume of 5 μL of isolated EVs was transferred to formvar-carbon-coated copper electron microscopy grids following the protocol of Thery et al. [91], with modifications. First, 5 μL of sample were mixed with 5 μL of paraformaldehyde (4%). The sample was transferred to the grid for 20 min and washed on a PBS drop. The samples were fixed by putting the grid on a glutaraldehyde (1.5%) drop for 5 min, washed several times in distilled water drop, and transferred to 0.5% of uranyl oxalate (Electron Microscopy Sciences, Hatfield, PN, USA) (pH 7.0) drop for 5 min to contrast the grid. Then, the grid was transferred to a 4% uranyl acetate drop for 10 min and left to dry on filter paper. Finally, each grid was placed on the TEM sample support to perform the search and acquisition of EV images at a magnification of 40,000× to 80,000× in the Talos™ F200C transmission electron microscope (Thermo Fisher Scientific). Approximately 5 images per sample were captured and processed with ImageJ software (V1.47t, NIH, Bethesda, MD, USA), the link to which is available at electronic address https://imagej.nih.gov/ij/download.html, accessed on 25 December 2021.

### 4.6. Endometrial Bovine Cells Isolation and Culture

Samples from endometrial tissues were collected from 1 cow in the luteal phase of the estrous cycle, culled in a local slaughterhouse. The reproductive phase of the animals was first determined by the structures in the ovaries and confirmed by the serum quantification of progesterone and estradiol using the radioimmunoassay technique (Laboratory of Animal Physiology and Endocrinology, University of Concepción, Chile). One cow with serological values of 3.53 pg/mL of estradiol and 11.5 ng/mL of progesterone was selected for derivation of endometrial epithelial cell primary culture. The epithelial cells were isolated using a protocol modified from Masuda et al., [92]. Tissue samples were taken from the ipsilateral horn to the corpus luteum. The obtained endometrial samples were processed under mechanical disruption of the tissues and subsequently by enzymatic action. Each tissue was collected in tubes for enzymatic digestion (1 mg/mL of collagenase (Colagenase type II, Gibco, Billings, MT, USA) plus 0.5 mg/mL of dispase (Dispase II, Gibco)) for 2 h at 37 °C with stirring. The resulting endometrial tissue fragments were separated into two fractions using sterile gauze; the first resulting fraction was filtered a second time using a 40 μm diameter filter (Corning^®^ cell strainer, Sigma Aldrich, St. Louis, MO, USA), and the residual cells generated in the second filtrate were considered to be pieces of glandular epithelium, which were seeded in 60-mm culture dishes in culture medium (Bovine Endometrial Cell Growth Medium, Sigma, St. Louis, MO, USA). The culture was checked on Day 2 post-seeding, and the stromal cells present in the cultures were cleaned with 0.25% trypsin (Trypsin-EDTA (0.25%), Gibco). Finally, all samples were cryopreserved in liquid nitrogen until later use.

The endometrial epithelial cells were characterized based on the positivity to cytokeratin (Dako Omnis clone AE1, number GA053) marker and negativity to vimentin (Dako Omnis, clone V9, number GA630), a stromal cell marker (Appendix A). For the immunocytochemistry, cells were treated and fixed in PFA 4% in PBS 15 min to room temperature. Samples were washed three times in PBS 1× for 5 min each, then block buffer was added for 60 min at room temperature. Primary antibody was added (vimentin 1:100 and cytokeratin 1:100) and then an isotype control IgG 1 (Jackson 1:100) was incubated for 2 h to prevent signals from unspecific unions. Then, the samples were washed with impress universal reagent (Vector laboratories) for 1 h at room temperature in the dark and placed in a coverslip slide with the addition of faramount aqueous mounting medium (Dako, Glostrup, Denmark).

### 4.7. EVs Internalization by Epithelial Endometrial Cells

For the experiments, the cells isolated above were cultured in 60 mm-diameter culture dishes using Bovine Endometrial Cell Growth Medium (B911-500, Merck, Rahway, NJ, USA) until they reached the confluence. Then, cells were trypsinized and seeded in 35 mm-diameter culture dishes at a density of 0.3 × 10^6^ until they reached the 60% confluence. At this time, the culture medium was replaced by an EV-depleted medium which was produced by ultrafiltration (centrifugal filter devices 100 kDa, Amicon, Miami, FL, USA) of culture medium (B911-500, Merck, Rahway, NJ, USA) for 15 min at 1660 g [93]. The EVs derived from in vivo- or in vitro-produced embryos were labeled with PKH67 (Sigma-Aldrich) fluorescent dye according to the manufacturer’s instructions to track the internalization. A total of 1 mL diluent C and 4 µL of PKH67 were added to each of the samples, which were then incubated at room temperature for 5 min. BSA (1%) was added to the samples; each sample was loaded to an Amicon^®^ Ultra-15 filter device (10 kDa; Merck, Rahway, NJ, USA) and centrifuged at 3000 rpm for 15 min. The samples in the filter were then washed 3 times using PBS and centrifuged at the same speed and for the same duration. The samples were recovered from the filter and quantified by NTA. A similar number of EVs (a volume containing 1 × 10^8^ EVs) was added to each cell culture dish. The number of EVs was selected based on previous report from Koh et al. [94]. An equivalent volume of PBS subjected to the same PKH67 labeling protocol was added to the cell culture dishes as a negative control. The internalization of the EVs was evaluated 24 h after supplementation. Cell nuclei were visualized using 2 drops per mL of DAPI dye (NucBlue^®^, Thermofisher Scientific, Santiago, Chile) and incubated for 15 min. Fluorescence was observed on the EVOS FL Imaging System (ThermoFischer Scientific, Santiago, Chile). Cells were kept in culture for another 24 h and then harvested for total RNA isolation.

### 4.8. Endometrial Epithelial Cell mRNA Sequencing

Total RNA was isolated from endometrial epithelial cells that were incubated with EVs from embryos or PBS. For this, the E.Z.N.A Total RNA Kit I (OMEGA, Atlanta, GA, USA, EEUU) was used following the manufacturer’s instructions. RNA quality control and quantification were carried out using the Agilent RNA Pico chip kit. From each sample, half of the total RNA was used for sequencing, while the other half was kept for real-time PCR. RNA sequencing was outsourced to CD Genomics (Shirley, NY, USA). The mRNA was obtained by using the NEBNext^®^ Poly (A) mRNA Magnetic Isolation module. Subsequently, the mRNA was fragmented, and the cDNA was synthesized, repaired, and ligated using the NEBNext^®^ Ultra™ II Directional RNA Library Prep Kit (New England Biolabs, Ipswich, MA, USA). The libraries were sequenced using the NextSeq 500/550 High Output V2 kit on the Illumina NextSeq 500 equipment (Illumina Inc., San Diego, CA, USA). Individual reads of 150 base pairs in length were obtained.

### 4.9. Bioinformatic Analysis

The quality of the resulting libraries was evaluated using the FastQC program [48] in combination with the VEC screen platform [95] to determine the presence of possible adapter sequences in the reads. The RNAQC chain software [96] was used to evaluate remnant sequences for possible prokaryotic RNA contamination. The readings were cut and filtered using the Trimmomatic software [97]. Sequences with a value over 30 Phred were selected. The length of mRNAs was accepted to be over 75 nucleotides. The sequences were mapped against the Bos Taurus reference genome (ARS-270 VCD1.2) using the HISAT2 program [98]. The SAM files obtained were converted to BAM files, which were sorted and indexed using SAM tools [99]. Identification of the transcripts was conducted using the StringTie package [100], and reads were expressed as kilobase per million (RPKM). Differential expression analysis was performed using the R package EdgeR [101], and only the genes with counts of at least 5 CPM were considered for this analysis. This analysis was performed using the TMM method and adjusted to the generalized linear binomial model. Genes were considered to be differentially expressed with the log2-fold change of more than 0.5 or less than −0.5 and when FDR (false discovery rate) <0.05. The results were plotted using the ggplot viewer on the heatmap, PCA, and volcano plot graphs. An enrichment analysis of molecular pathways was performed with the differentially expressed genes, based on the KEGG database, and visualized using the ClusterProfiler package [102].

### 4.10. Quantitative Reverse Transcription PCR Analysis

To validate the RNAseq data, the expression pattern of 13 selected genes was analyzed by real-time PCR (RT-PCR). A total of 500 ng of RNA obtained from epithelial endometrial cells was used to produce complementary DNA (cDNA). The total RNA was treated with 1 U of RNAse-free DNase I (Invitrogen, Waltham, CA, USA) for 30 min at 37 °C to remove genomic DNA contamination. DNase was inactivated by adding 25-mM EDTA (1 μL) to each sample and incubated at 65 °C for 10 min. Then, 1 μL of random primers (150 ng/mL) and 10 mM of each deoxynucleotide triphosphate (dNTP) were added and the samples were incubated at 65 °C for 5 min. Later, 4 μL of M-MLV RT buffer (5× first strand buffer vitrogen™), 0.1 M of dithiothreitol (DTT) and 40 U/μL of RNAse inhibitor (RNAsa out) were added. The next incubation was performed at 37 °C for 2 min. Finally, 200-U/μL of the M-MLV Reverse Transcriptase (Thermo Scientific) were added and samples were incubated at 25 °C for 10 min, 37 °C for 50 min, and 70 °C for 15 min. The obtained cDNA was stored at −20 °C for later use.

The RT-PCR reaction was performed with 2 μL of cDNA, 10 pmol from each primer (forward and reverse), and 5 μL of the master mix from Kicq Start Syber green qPCR (Sigma, St. Louis, MO, USA) kit, creating a final volume of 10 μL. Each sample was loaded in duplicate (technical replicates). Primer details are listed in Table 5. The primers were made using the Amplifx program and the NCBI database. The reaction was run on a MX3000P Real-Time PCR device (Agilent, Santa Clara, CA, USA). Gene expression analysis was performed using the 2^−ΔΔCT^ method. For each sample, a CT (cycle threshold) value was normalized to the geometric mean of ACTB (Beta-actin) and GAPDH (Glyceraldehyde-3-phosphate dehydrogenase). Statistical analysis was conducted using the Kruskal Wallis test. Validation of RNAseq was conducted by Pearson’s correlation test with the log2 of fold change. All the analyses were performed using the R package EdgeR. Significant differences were determined when *p* < 0.5.

## 5. Conclusions

In this work, it was confirmed that the EVs released by bovine embryos from Day 5 to Day 7 of development are internalized by epithelial endometrial cells, inducing the modification of their gene expression pattern. Several pathways are affected by the embryonic EVs that are of great importance to the regulation of the expression of genes associated with endometrial function and IFNT signaling. In addition, different embryo origins, in vivo or in vitro, generate different EVs, which impact the ability of said EVs to modify the gene expression patterns in the recipient cells. This work represents evidence of the role of the extracellular vesicles in embryo-maternal communication and how the embryo quality could affect this interaction.

## Figures and Tables

**Figure 1 ijms-24-07438-f001:**
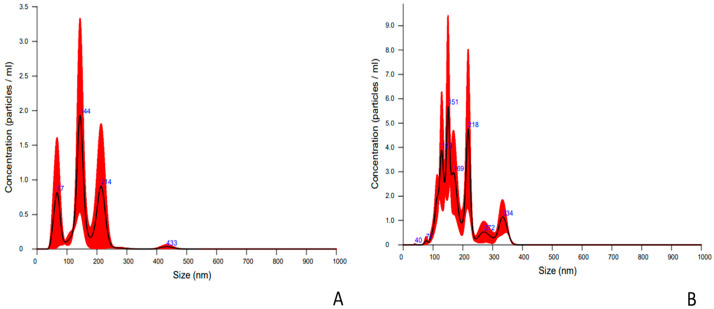
Size distribution (X-axis) and concentration (Y-axis; (10^9^ particles × mL)) of EVs released by bovine embryos produced in vitro (**A**) or in vivo (**B**) during blastulation (morula-blastocyst). The graphics represent the average of three technical replicates, based on the diameter and concentration distribution of nanoparticles.

**Figure 2 ijms-24-07438-f002:**
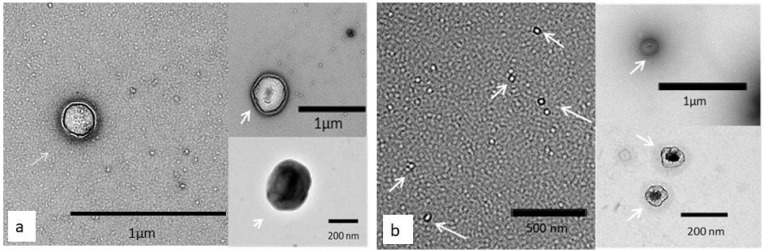
EVs secreted by bovine embryos during blastulation (morula to blastocyst). (**a**,**b**): representative images of EVs isolated from culture medium of embryos produced in vivo and vitro, respectively. Arrows are pointing identified EVs in the representative pictures.

**Figure 3 ijms-24-07438-f003:**
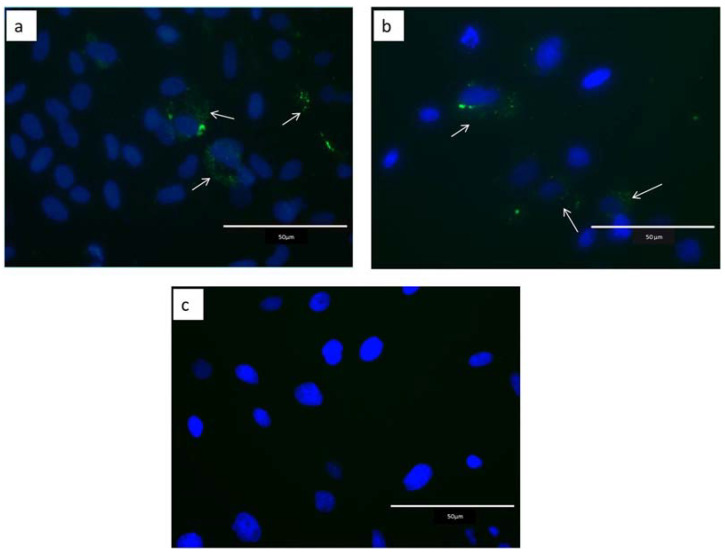
Fluorescence images of epithelial endometrial cells incubated with EVs secreted during the blastulation of bovine embryos produced in vivo (EVs-IVV) or in vitro (EVs-IVP). (**a**) EVs released from embryos produced in vitro. (**b**) EVs released from embryos produced in vivo. (**c**) Negative control (PBS/PHK67). Cells were incubated with stained vesicles for 24 h. Arrows highlight the green dots in the cell cytoplasm, indicating internalized EVs stained with PHK67 dye.

**Figure 4 ijms-24-07438-f004:**
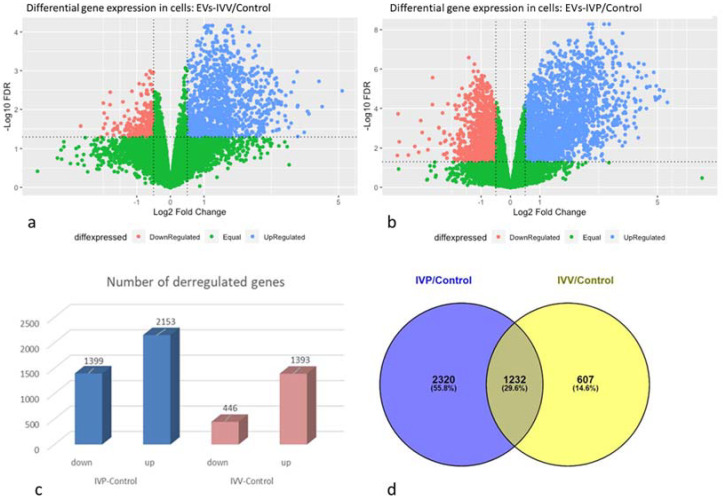
Analysis of deregulated genes in bovine epithelial endometrial cells due to the effect of extracellular vesicles released during blastulation by bovine embryos produced in vivo (IVV) or in vitro (IVP). (**a**,**b**): Volcano plot showing the comparison between EVs from IVV embryos and PBS (negative control) and between EVs from IVP embryos and PBS (negative control), respectively. (**c**): Number of deregulated genes (up or down), comparing EVs from in vitro produced embryos with PBS (IVP-control), and EVs from in vivo produced embryos with PBS (IVV-control). (**d**). Venn diagram of differentially expressed genes from the analyses. Venn diagram was constructed using Venny 2.1 https://bioinfogp.cnb.csic.es/tools/venny/; accessed on 6 December 2022.

**Figure 5 ijms-24-07438-f005:**
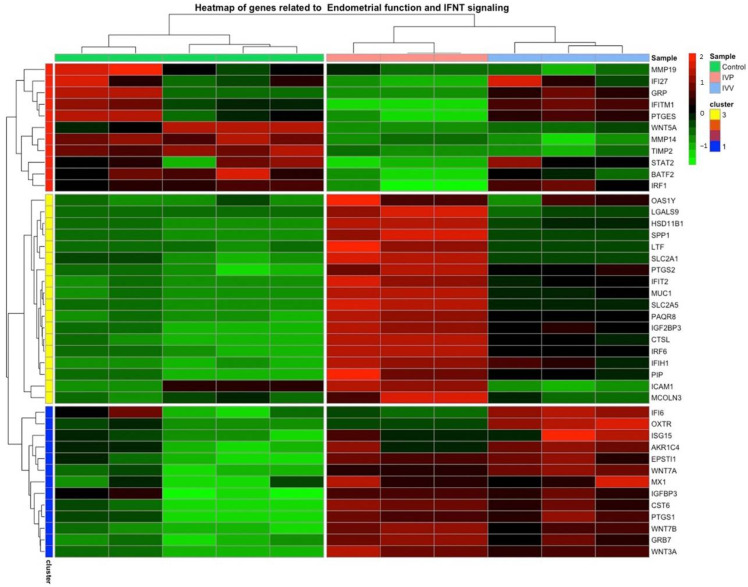
Heat map profile of differentially expressed genes related to IFNT signaling and endometrial function. Three clusters were defined: Cluster 1 includes genes associated with sex steroid signaling and interferon signaling; Cluster 2 includes solute and water transport secretory activity, proliferation activity, and interferon signaling; Cluster 3 includes extracellular matrix remodeling, proliferation activity and interferon signaling.

**Figure 6 ijms-24-07438-f006:**
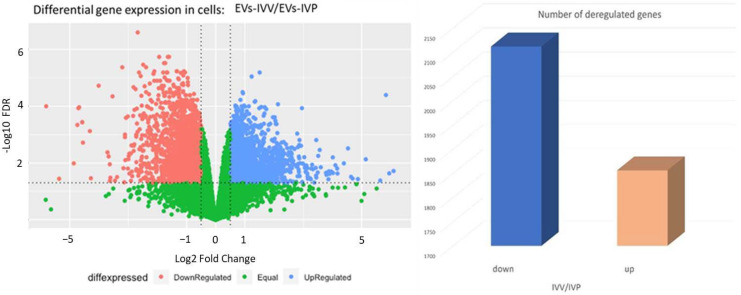
Effect of extracellular vesicles (EVs) secreted by bovine embryos produced in vivo (IVV) or in vitro (IVP) on the gene expression pattern of epithelial endometrial cells.

**Figure 7 ijms-24-07438-f007:**
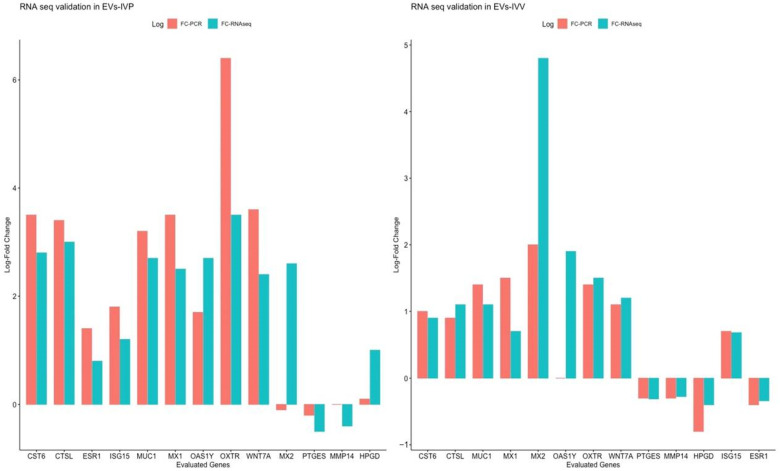
Fold change graphic comparing the fold change in RNA seq and RT qPCR of each evaluated gene. Data on Y-axis represent the log of fold change obtained from PCR analysis (red) and RNAseq (green). A Pearson correlation analysis showed a similar fold change tendency for most of the genes except *MX2* in the EVs-IVP group.

**Figure 8 ijms-24-07438-f008:**
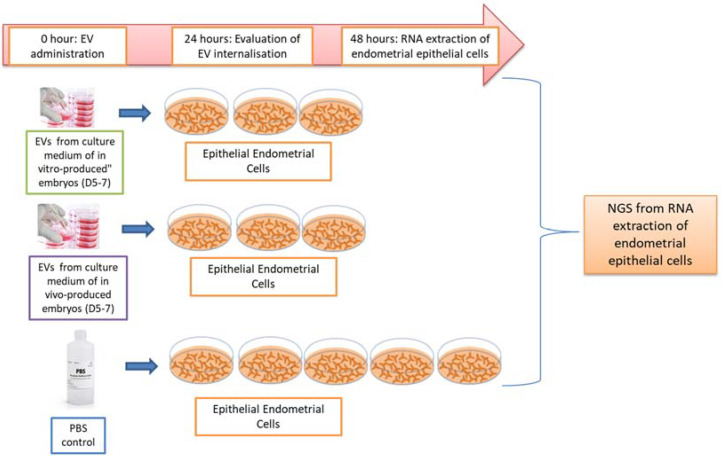
Illustration of experimental design. Extracellular vesicles (EVs) were isolated from a culture media of in vitro (IVP)- or in vivo (IVV)-derived bovine embryos. Morulae were selected on Day 5 of development and cultured individually until Day 7. Culture media were collected from embryos that developed to blastocyst stage. Isolated EVs were stained with PKH67 and added to in vitro-cultured bovine endometrial cells to assess EV internalization 24 h later. Cells were harvested 48 h after the addition of EVs for gene expression analysis. Cells treated with PBS were used as a negative control.

**Table 1 ijms-24-07438-t001:** Size and concentration of EV populations secreted during blastulation by bovine embryos produced in vitro or in vivo.

EVs Parameter	EVs from In Vivo EMBRYOS	EVs from In Vitro Embryos
Media, nm (x +/− sd)	186.4 +/− 62.9 nm	168.9 +/− 80.4 nm
Mode, nm (x +/− sd)	149.2 +/− 11.2 nm	133.7 +/− 10.0 nm
Concentration	4.24 × 109 +/− 4.36 × 10^8^ particles/ml	1.14 × 109 +/− 8.62 × 10^8^ particles/mL ^1^

^1^ Sd. Standard deviation; EVs: extracellular vesicles. No statistical analysis was performed.

**Table 2 ijms-24-07438-t002:** Evaluation of surface markers CD63, CD40, CD9, and CD81 in EVs secreted during blastulation by bovine embryos produced in vitro or in vivo.

Sample	CD63	CD81	CD40	CD9
5–7 in vivo	3%	9.4%	25.7%	9.8%
5–7 in vitro	7.1%	5.4%	55%	9.5%
Bovine Blood Serum (Positive control)	5.6%	8.1%	14%	15.4%
Beads (negative control)	0%	0%	0%	0%
Beads + antibody (negative control)	0%	0%	0%	0%
SOFdep (Oviductal synthetic fluid depleted)	0.78%	1.98%	0.74%	0.92% ^1^

^1^ EVs-IVV 5–7 and EVs-IVP 5–7: EVs released during blastulation (Days 5–7 of development) by bovine embryos produced in vivo or in vitro, respectively. SOFdep: Oviductal synthetic fluid depleted of EVs. Culture medium without contact with embryos was used as negative control.

**Table 3 ijms-24-07438-t003:** Genes related to endometrial function and IFNT signaling with significant differences between EVs-IVV and EVs-IVP treatment.

Gene ID	Category	Log2FC IVV/IVP	Reference
*BATF2*	Bovine endometrium	1.10	[41]
*CTSL*	Stimulated by IFNT Ovine endometrium	−0.95	[42,43]
*GRP*	Stimulated by IFNT in ovine uterus	2.19	[44]
*HSD11B1*	Stimulated by IFNT Ovine endometrium	−1.39	[42,43]
*ICAM1*	Bovine endometrium	−1.5	[8]
*IFI6*	Induced by IFNT in bovine uterus	0.87	[45]
*IRF1*	bovine endometrium	1.07	[41]
*IRF6*	Induced by IFNT Ovine endometrium	−0.85	[46]
*LGALS9*	Bovine endometrium	−2.51	[45]
*LTF*	Bovine endometrium	−2.2	[47]
*MCOLN3*	Bovine endometrium	−1.4	[8]
*MUC1*	Bovine endometrium	−1.09	[8]
*PAQR8*	Bovine endometrium	−0.68	[48]
*PIP*	Bovine endometrium	−0.97	[49]
*PTGES*	Corpus luteum and endometrium function	0.95	[50]
*SLC2A1*	Stimulated by IFNT Ovine endometrium	−0.93	[42,43]
*SPP1*	Stimulated by IFNT Ovine endometrium	−1.33	[42,43]
*STAT2*	Ovine uterus	0.52	[51] ^1^

^1^ log2FC: log2 of the mean ratios of expression. Genes were considered to be differentially expressed when log2FC > 0.5 or <−0.5 and FDR ˂ 0.05. References indicate reports linking the genes to the function or signaling pathway.

**Table 4 ijms-24-07438-t004:** Genes associated with IFNT signaling or endometrial function that are significantly deregulated in epithelial endometrial cells due to the effect of extracellular vesicles secreted by bovine embryos produced in vivo (EVs-IVV) or in vitro (EVs-IVP).

Gene ID	Category	Log2FC EVs-IVV/Control	Log2FC EVs-IVP/Control	Reference
*OAS1Y*	INFT	1.4	2.55	[10]
*MX1*	INFT	0.70	2.48	[10]
*ISG15*	INFT	0.68	1.23	[10]
*MX2*	INFT	4.79	2.62	[10]
*IRF6*	INFT	1.42	2.28	[46]
*IFIH1*	INFT	1.02	1.53	[52]
*HSD11B1*	INFT	1.12	2.51	[53]
*CTSL*	INFT	1.68	2.64	[53]
*CST6*	INFT	1.87	2.12	[53]
*MUC1*	Cell adhesion	1.37	2.47	[8]
*LGALS9*	Cell adhesion	1.25	3.77	[45]
*WNT7B*	Early embryonic development and endometrial function	2.05	2.39	[54]
*WNT7A*	Early embryonic development and endometrial function	2.14	1.81	[53]
*WNT5A*	Early embryonic development and endometrial function	−0.70	−0.95	[55,56]
*WNT3A*	Early embryonic development and endometrial function	2.81	3.15	[55]
*PTGS2*	Endometrial function	0.58	1.06	[53]
*PTGS1*	Endometrial function	2.72	2.77	[57]
*MMP19*	Extracellular matrix remodeling	−0.61	−0.5	[8]
*GRB7*	Growth factor signaling	1.49	1.86	[58]
*IGF2BP3*	Growth factor signaling	1.14	1.77	[59]
*IGFBP3*	Growth factor signaling	1.08	1.15	[60]
*IFIT2*	Immune response	1.43	2.47	[45]
*EPSTI1*	Immune response	0.8	0.7	[45]
*SPP1*	Secretory activity	0.58	1.91	[42,43]
*PIP*	Secretory activity	2.27	3.25	[49]
*LTF*	Secretory activity	0.95	3.16	[47]
*OXTR*	Sex steroid signaling	1.64	3.43	[61]
*PAQR8*	Sex steroid signaling	0.89	1.78	[48]
*AKR1C4*	Sex steroid signaling	0.59	4.14	[8]
*ESR1*	Sex steroid signaling	−0.34 (not significant)	0.77	[8] ^1^

^1^ log2FC: log2 of the mean ratios of expression. Genes were considered to be differentially expressed when log2FC > 0.5 or <−0.5 and FDR ˂ 0.05. References indicate reports linking the genes to the function or signaling pathway.

**Table 5 ijms-24-07438-t005:** Details of primers used for RT-PCR analysis of selected genes for validation of RNAseq data.

Gene	Primer Sequences (5′–3′)	Product Length (bp)	Annealing Temperature	Accession Number (NCBI)
*MX2*	F: AAGTATGAGGAGAAGGTGCGGC	112	57 °C	XM_015473641.2
	R: AGCTCTGGTCCCCGATAACG			
*OXTR*	F: ACAAGCACTCGCGCCTCTTCT	107	56 °C	XR_003031707.1
	R: GCGGAACGTGATGTCCCACAGA			
*MUC1*	F: ACATCCAGGCCCCTTTCCTC	118	55 °C	NM_174115.3
	R: GTGGAAACTGGCGTAGTTCTGC			
*OAS1Y*	F: AGCACCGTGATGGAGCTCAG	111	55 °C	NM_001040606.1
	R: GTCGATGGCTTCTTTGACCTGC			
*CTSL*	F: CCTCGCCACAGGTTTTTGAAC	91	55 °C	NM_001083686.2
	R: TGTGGTCAAATTTTGGAGCAGC			
*WNT7A*	F: ATGGTCTACCTCCGGATCGGTG	83	57 °C	NM_001192788.1
	R: GCCAGGCCTGGGATCTTGTTAC			
*ISG15*	F: ACCTGACGGTGAAGATGCTAGG	98	57 °C	NM_174366.1
	R: GATCTTCTGGGCGATGAACTGC			
*ESR1*	R: CCAACAGGTGCCCTATTACCTG	108	55 °C	NM_001001443.1
	R: CCACCTTGGCGTCGATTATCTG			
*HPGD*	F: AAGGTAGCGCTGGTCGATTGG	120	54 °C	NM_001034419.2
	R: TTGTTCCTGATCGGCCACATCG			
*PTGES*	F: ATGTACGTGGTGGCCGTCATC	108	56 °C	NM_174443.2
	R: GTCGTTCCGGCAATACTGGAGG			
*MMP14*	F: AGTCCCTCTCAGCTGCCATTG	89	56 °C	NM_174390.2
	R: CATGGCCTTCATGGTGTCTGC			
*MX1*	F: AGGCAGGAGACCATCAACTTGG	144	56 °C	XM_015473623.2
	R: ACCTTTGTCCACCAGATCGGG			
*CST6*	F: GTGTCCCTGTAGAGACCCCT	113	55 °C	NM_001012764.3
	R: TGCCGTAGATACGGTCCAAG			
*ACTB*	F: TGCCCTGAGGCTCTCTTCCA	119	55 °C	NM_173979.3
	R: TTGGCGTAGAGGTCCTTGCG			
*GAPDH*	F: AGGTCGGAGTGAACGGATTC	85	55 °C	NM_001034034.2
	R: ATGGCGACGATGTCCACTTT ^1^			

^1^ Abbreviation: F: forward; R: reverse; RT-PCR: real-time polymerase chain reaction.

## Data Availability

Data from RNAseq was uploaded to NCBI’s Sequence Read Archive (SRA) with accession code PRJNA931759.

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
