# Peer review of "Extracellular Vesicles Secreted by Pre-Hatching Bovine Embryos Produced In Vitro and In Vivo Alter the Expression of IFNtau-Stimulated Genes in Bovine Endometrial Cells"

_ijms, 2023, doi:10.3390/ijms24087438_

Round 1
Reviewer 1 Report
In the present manuscript, the authors demonstrate that extracellular vesicles (EVs) secreted by bovine embryos during blastulation (D5-D7) could induce transcriptomic modification, activating IFNT signaling in endometrial cells. This would be of potential importance by providing evidence regarding the effect of embryo origin (in vivo or in vitro) on the early embryo-maternal interaction mediated by extracellular vesciles, but there are several deficiencies that need to be addressed or clarified for further improvement of the manuscript.
Major
1. In Figure 2A, do the three peaks represent three technical replicates? If so, why are there huge difference in size?
2. Figure 4 A and B, please adjust the brightness. Please stain for the surface marker to demonstrate the intake of PHK67-labeled EVs.
Minor
1. The figure and table number is not in order.
2. Figure 3 legend, there is an extra period. Please revise accordingly.
3. Figure 2A, why are there two figures of scale 1 um?
Author Response
In response to comments from reviewers
Reviewer 1
- In Figure 2A, do the three peaks represent three technical replicates? If so, why are there huge difference in size?
Figure 2A represents the mean of three technical replicates from a sample. Each peak does not represent individual technical replicates. The image is the histogram from NTA analysis based on the nanoparticle distribution in terms of size (X) and concentration (Y).
- Figure 4 A and B, please adjust the brightness. Please stain for the surface marker to demonstrate the intake of PHK67-labeled EVs.
The brightness of this figure was adjusted as suggested by the reviewer and included in the revised version of this manuscript.
Regarding the second comment of the reviewer, we understand the concern and appreciate the suggestion. However, at the moment it is impossible to perform a new picture with a different staining since the complete experiment would have to be performed. For this experiment, we used EVs from in vivo and in vitro produced embryos. A high number of embryos were produced to collect the necessary amount of EVs to perform the incubation with cells for the transcriptomic analysis. Nevertheless, we indicate that EVs are internalized because the green dots were localized very closed and around to the nucleus. No green dots were observed in the control group. But we consider that the most important indication of EVs internalization was the transcriptomic changes that were detected in cells exposed to EVs from embryos produced in vivo or in vitro.
Minor comments
- The figure and table number is not in order.
This was corrected in the current version of the manuscript
- Figure 3 legend, there is an extra period. Please revise accordingly.
This was corrected in the current version of the manuscript
- Figure 2A, why are there two figures of scale 1 um?
The figure 2 are composed by two subfigures (A,B). The data are showing a histogram where the x-axis denotes the unit scale in nanometers being 1000 nm or 1um, the maximum value registered by the equipment
Reviewer 2 Report
This study looks at the effects of early embryo derived extracellular vesicles (EVs) on gene expression in endometrial cells. In addition, it seeks to look at whether in vitro (IVP) versus in vivo (IVV) embryo derived EVs have differential effects on the endometrial cell transcriptome. The authors claim that the effect of EVs on gene regulation from these EVs on the endometrium is unknown. This may be true, however, early embryo EVs in humans are known to interact with the endometrium and deliver RNA and regulate gene expression. In addition, it is already well established that IVP vs IVV and actually SCNT (somatic cell nuclear transfer) embryos have a differential effect on endometrial gene expression profile in ruminants. Therefore, the advance in this paper is just putting together two observations that are already known, although not investigated together. Since the effects are on cultured endometrium, it adds another layer of complexity to the results. It is always hard to assess cultured cells and their relevance although the field does use cultured cells for these purposes. Given these observations, I do not think there is such a big advance here. It is interesting to see the GO analysis but I feel the data needs to be substantiated more. A major concern is noted below:
Major comment:
In figure 4, there are only a few cells that show uptake of the EVs, which is not surprising since this is in culture. Since the RNA-seq is not a single cell experiment, but on pooled samples, how are the authors getting such consistent and large differences that they say have low variability? I would think that with such few cells and each cell only having a couple of log fold change, the cells that are not changed would dominate the RNA-seq reads. Also, if the cDNA comes from a pool of cells where most of them (as shown in the picture) do not contain EVs, then quantitative PCR should not show such a large change or consistent change. How many times was this experiment done? I would expect that the change would differ every time depending on how many cells received EVs and responded to it, which would be different in each experiment. Given that, the entire paper is based on these observations, if they are not credible, the paper cannot stand. Explanation of the results in terms of the expected abundance of the differential transcripts should be explained in the context of figure 4. A very useful number would be what proportion of cells in culture take up EVs and how consistent is that? Are the EV positive and negative cells separated? With that number we could evaluate how likely it is to give consistent results in terms of RNA-seq or q-PCR. Otherwise the invariable consistent changes are hard to grasp.
Author Response
In response to comments from reviewers
Reviewer 2
Major comment:
- In figure 4, there are only a few cells that show uptake of the EVs, which is not surprising since this is in culture. Since the RNA-seq is not a single cell experiment, but on pooled samples, how are the authors getting such consistent and large differences that they say have low variability? I would think that with such few cells and each cell only having a couple of log fold change, the cells that are not changed would dominate the RNA-seq reads. Also, if the cDNA comes from a pool of cells where most of them (as shown in the picture) do not contain EVs, then quantitative PCR should not show such a large change or consistent change. How many times was this experiment done? I would expect that the change would differ every time depending on how many cells received EVs and responded to it, which would be different in each experiment. Given that, the entire paper is based on these observations, if they are not credible, the paper cannot stand. Explanation of the results in terms of the expected abundance of the differential transcripts should be explained in the context of figure 4. A very useful number would be what proportion of cells in culture take up EVs and how consistent is that? Are the EV positive and negative cells separated? With that number we could evaluate how likely it is to give consistent results in terms of RNA-seq or q-PCR. Otherwise the invariable consistent changes are hard to grasp.
We understand the reviewer's concern and appreciate the point that it is indeed a wonderful idea if one wants to understand the effect of EVs on a cell and not on the overall response as we intended in this paper. We agree that only few green dots, indicative of the presence of EVs were observed. However, we based the experimental design on a reference, adjusting the number of nanoparticles to the number of cells in culture as recommended, hence the selection of this model was not random and has been proposed by other authors. For each experimental groups, three replicates were independently performed, obtaining the low variability showed in the PCA analysis (supplementary figure 2). We believe that the response that was observed in the cells, treated with EVs must be due to the EVs effect because was quite similar in all replicates. The image in figure 4 is representative of a section of the culture dish and was taken 24 hrs after adding the EVs to the culture, it is possible than only few EVs are visualized at this moment. Also, it is important to remark that considering the size of EVs, some of them could probably “escape” from the visual range of the microscopy. The control group was handled in the same way than the experimental groups, a volume of PBS was treated with PKH67 and added to the cells in culture, not green spots were detected in these cells and their transcriptomic profile is statistically different. The bioinformatics team performed a robust analysis ending up with a list of differentially expressed genes with an FDR of 0.05, demonstrating differences among groups: embryonic EVs vs control but also between EVs released by embryos produced in vivo and in vitro.
We also would also would like to mention that a single cell analysis would be interesting to have an idea of how the EVs directly affect a cell, however this will not represent what might happen in an in vivo system. We believe that a similar behavior will occur in vivo considering the amount of EVs that could reach the endometrial cell and how strong the response could be. However, this response could be amplified during the time of elongation in vivo, having a stronger effect on embryo-maternal interaction to ensure the implantation. In another works, (Sponchiado et al) had been demonstrated the effect of the embryos on endometrial cells as a consequence of the proximity of the embryo or the gradient of secreted molecules. In this work, we intended to demonstrate that embryos are able to produce signals since very early in development that will have an affect (probably still weak) on the maternal cells. Our hypothesis also pointed to the EVs as a relevant mechanism participating in this early interaction and in our opinion, this was demonstrated.
Round 2
Reviewer 1 Report
The authors have successfully addressed my comments. I don't have further comments.
Reviewer 2 Report
Response is satisfactory. This explanation in brief should be added to guide readers.